# Dynamic Gradient Influencing for Viral Marketing Using Graph Neural Networks

## Abstract

The problem of maximizing the adoption of a product through viral marketing in social networks is of extreme importance and has been studied heavily through postulated network models. We present a novel data-driven formulation of the problem. We use Graph Neural Networks (GNNs) to model the adoption of products by utilizing both topological and attribute information. The resulting *Dynamic Viral Marketing (DVM)* problem seeks to find the minimum budget and minimal set of dynamic topological and attribute changes in order to attain a specified adoption goal. We show that DVM is NP-Hard and is related to the existing influence maximization problem. Motivated by this connection, we develop the idea of Dynamic Gradient Influencing (DGI) that uses gradient ranking to find optimal perturbations and targets low-budget and high influence non-adopters in discrete steps. We use an efficient strategy for computing node budgets and develop the "Meta-Influence" heuristic for assessing a node's downstream influence. We evaluate DGI against multiple baselines and demonstrate gains on average of 24% on budget and 37% on AUC on real-world attributed networks. Our code will be made publicly available.

## CCS Concepts

• **Do Not Use This Code → Generate the Correct Terms for Your Paper**; *Generate the Correct Terms for Your Paper*; Generate the Correct Terms for Your Paper; Generate the Correct Terms for Your Paper.

## Keywords

Do, Not, Us, This, Code, Put, the, Correct, Terms, for, Your, Paper

**ACM Reference Format:**
Anonymous Author(s). 2018. Dynamic Gradient Influencing for Viral Marketing Using Graph Neural Networks. In *Proceedings of Make sure to enter the correct conference title from your rights confirmation emai (Conference acronym 'XX)*. ACM, New York, NY, USA, 12 pages. https://doi.org/XXXXXXX.XXXXXXX

## 1 Introduction

Viral marketing is a highly significant strategy used to maximize the adoption of products [7, 25], through diffusion in a social network of users [22]. Prior work is based mainly on postulated network propagation models of viral phenomena that focus on static topologies and ignore node attributes [14]. Furthermore, while finding the influential seed set has been extensively studied, the problem of making "dynamic" topological and attribute perturbations to maximize spread from adopters to non-adopters has not been addressed.

Instead of designed/postulated network propoagation models, we adopt data-driven models, specifically non-linear Graph Neural Networks (GNNs) [6, 15] to learn a propagation model directly from attributed network data, and then use it to forecast future states of the spread after the network is perturbed. We train a GNN model on the initial state of the attributed network to learn a data-driven mapping from user attributes and neighborhood to its adoption label. Thereafter, the GNN parameters are fixed and the decision boundary of the GNN is used to identify adopters and non-adopters after the network is perturbed. This *self-labeling* technique allows us to study the effect of perturbations on user adoption by alleviating the issue of data scarcity regarding users with unseen combinations of attribute and neighborhoods.

In order to model the effect of perturbations, we propose a realistic model that can be used to strategically accelerate spread from adopters to non-adopters. The attributed networks we consider are unweighted, undirected graphs with binary node attributes and labels, where states of both attributes and labels correspond to the adoption of marketable products. Accordingly, at any given time, (a) new edges can be added only between adopters and non-adopters, as in referral marketing [2], and (b) adopters can further adopt similar products or products by flipping corresponding attributes from 0 to 1, as in joint or co-marketing [12].

The resulting Dynamic Viral Marketing (DVM) problem seeks to find the minimum budget and minimal dynamic perturbation set to attain a spread goal. We show that DVM is NP-Hard and relate it to the Influence Maximization (IM) problem [14]. Similar to IM under the linear threshold model, nodes in DVM flip when the sum of incoming influence edge weights exceeds the node's adoption threshold. Despite the similarity, the two problems are different as incoming influence edge weights and node thresholds in DVM are dynamic and governed by the underlying GNN propagation model.

Motivated by the connection of IM to DVM, we develop the Dynamic Gradient Influencing (DGI) framework to solve the DVM problem. DGI unrolls in discrete steps; each step involves flipping a non-adopter node that has the lowest budget and maximum downstream influence. We use gradient-guided node flipping to find the required dynamic perturbations. We develop an efficient node flipping budget computation approach using bisection search to maintain node budgets at each step. To estimate a node's downstream influence, we develop the gradient based "Meta Influence" heuristic and the corresponding "Meta Attribute Flips" to increase the potency of edge perturbations.

Our contributions are as follows:

*Conference acronym 'XX, June 03–05, 2018, Woodstock, NY*
© 2018 Copyright held by the owner/author(s). Publication rights licensed to ACM.
ACM ISBN 978-1-4503-XXXX-X/18/06
https://doi.org/XXXXXXX.XXXXXXX

- We propose the novel Dynamic Viral Marketing (DVM) problem to find the minimum budget and a minimal dynamic perturbation set to attain a spread goal, where a non-linear GNN acts as the propagation model and perturbations are restricted by referral and co-marketing constraints. We show that DVM is NP-Hard and is connected to the influence maximization (IM) problem.
- We develop the Dynamic Gradient Influencing (DGI) framework that targets low budget and high influence non-adopters. DGI consists of (a) an efficient budget computation approach, (b) a novel Meta Influence heuristic and Meta Attribute Flips to increase node influence.
- We comprehensively evaluate DGI on three real-world attributed networks and demonstrate gains on average of 24% on budget and 37% on AUC over multiple baselines. Further, we extensively analyze the cascade patterns and intermediary adopter nodes discovered by DGI.

## 2 Related Work

*Models for Network Diffusion.* The modeling of the diffusion of innovations in a network through the process of social contagion is a long studied topic [16]. Granovetter [10] developed a threshold based model of collective behaviour where individuals are influenced by the proportion of others who come to a particular decision. Morris [22] studied a coordination game of direct benefits from aligning choices with neighbors in a social network. In epidemiology, the spread of biological disease is studied using probabilistic transmission models of Susceptible, Infected and Recovered (SIR) individuals [24]. The use of the network value of customers for marketing was first explored in [7, 25]. Consequently, Influence Maximization (IM), the problem of finding the most influential seed set for viral marketing has been studied extensively [9, 14, 30]. Dynamic Viral Marketing lies within the broad class of spreading network processes and we show how it is connected to IM.

*Graph Neural Networks.* Graph Neural Networks (GNN) are message passing neural networks that operate on attributed networks and have shown great success in problems such as node classification, link prediction, recommendation systems, and community detection. Various GNN architectures have been proposed since their first inception—Graph Convolutional Networks [15], GraphSAGE [11], Graph Attention Networks [28], Simplifying Graph Convolutional Networks [31]. We refer the reader to [32] for an extensive survey of graph neural networks. We use GNNs as the underlying propagation model for Dynamic Viral Marketing.

*Gradient-based Network Optimization.* Gradient-based network optimization is used in many combinatorial network optimization problems. In the context of adversarial attacks on GNNs, perturbations are made using gradient optimization on the input network strucure to reduce the accuracy of a GNN classifier [5, 33, 35]. Global attacks with dynamic budget adjustment for Topology PGD [33] are considered by [34]. In [20], reinforcement learning policies are optimized to solve Maximum Coverage, Vertex Cover, and Influence Maximization problems on networks. In [18], graph neural networks are optimized to create graph embeddings and predict influence of nodes for solving Influence Maximization. We use gradient-based network optimization within the DGI framework.

## 3 Preliminaries

Consider a graph $G = (A, X)$, with the associated adjacency matrix $A \in \{0, 1\}^{n \times n}$ and node attribute matrix $X \in \{0, 1\}^{n \times d}$ respectively, and node labels $Y \in \{0, 1\}^n$. We refer to the associated node-ids as $\mathcal{V} = \{1, \ldots, n\}$. We denote the node feature $x_v \in \{0, 1\}^d$, and the node label $y_v \in \{0, 1\}$. The set of adopters and non-adopters is denoted as $S$ and $D$ respectively. For convenience, we denote the sub-matrix of $A$ defining node connectivity from a set of nodes $S$ to a set of nodes $D$ as $A_{S,D}$, the sub-matrix of $X$ containing features for a set of nodes $S$ as $X_S$, and the vectors of ones and zeros as $\mathbf{1}$ and $\mathbf{0}$ respectively. We denote the weights on edge and feature perturbations as $P_A$ and $P_X$ respectively. The gradient scores on edge and feature perturbations are denoted by $\hat{P}_A$ and $\hat{P}_X$ respectively. For a dynamically changing network, the superscript $t$ is used to indicate the variable at time $t$; we drop the superscript if it is clear from the context.

We consider the large family of graph neural networks [11, 15] to construct layerwise hidden representations and finally output classifier logit scores $Z \in \mathbb{R}^{n \times 2}$. For an L-layer GNN,

$$H_l = \sigma(\hat{A} W_l H_{l-1}),$$
$$H_0 = X, \quad Z = H_L$$

where $W_l$ refers to the learnable GNN parameters at layer l, $\sigma$ is a nonlinear activation function, and $\hat{A}$ is the GNN propagation matrix. For Graph Convolutional Networks (GCN) [15], $\hat{A} = \bar{\Delta}^{-\frac{1}{2}} \bar{A} \bar{\Delta}^{-\frac{1}{2}}$, where $\bar{A} = A + I$ and $\bar{\Delta}$ is the associated degree matrix. For GraphSAGE [11] with mean pooling, $\hat{A} = \bar{\Delta}^{-1} \bar{A}$. The predicted labels $y'_v \in \{0, 1\}$ for each node $v \in \mathcal{V}$ are given by the class with the maximum logit score. In the typical semi-supervised learning scheme for node classification, referred to as transductive learning, the GNN parameters $W$ are learned by minimizing the cross-entropy classification loss,

$$\mathcal{L}_{tr}(v) = -\log \sigma(z_v)_{y_v} \tag{1}$$

where $z_v$ denotes logit scores for node $v$ and $\sigma$ the softmax function.

## 4 Dynamic Viral Marketing Problem

In this section, we present the problem of viral marketing [7, 25] in the context of dynamic changes for accelerating the adoption of products by customers. Specifically, we consider a dynamic marketing scenario with the following salient properties:

(1) *Referral marketing*: Companies incentivize people who arere already using their product to refer it to others; adopters make new connections in the network to non-adopters.

(2) *Co-marketing*: Companies partner with other companies or jointly market a group of products; people who adopt the target product likely adopt similar products and vice versa.

Formally, consider $G^t = (A^t, X^t, Y^t)$, a series of undirected and unweighted dynamic attributed networks, observed at discrete time steps $t = 1, \ldots, T$, where $A^t$ represents the adjacency matrix defining node connectivity, $X^t$ represents a binary node feature matrix containing adoption labels for related products, and $Y^t$ represents the binary adoption labels for the target product. We assume that

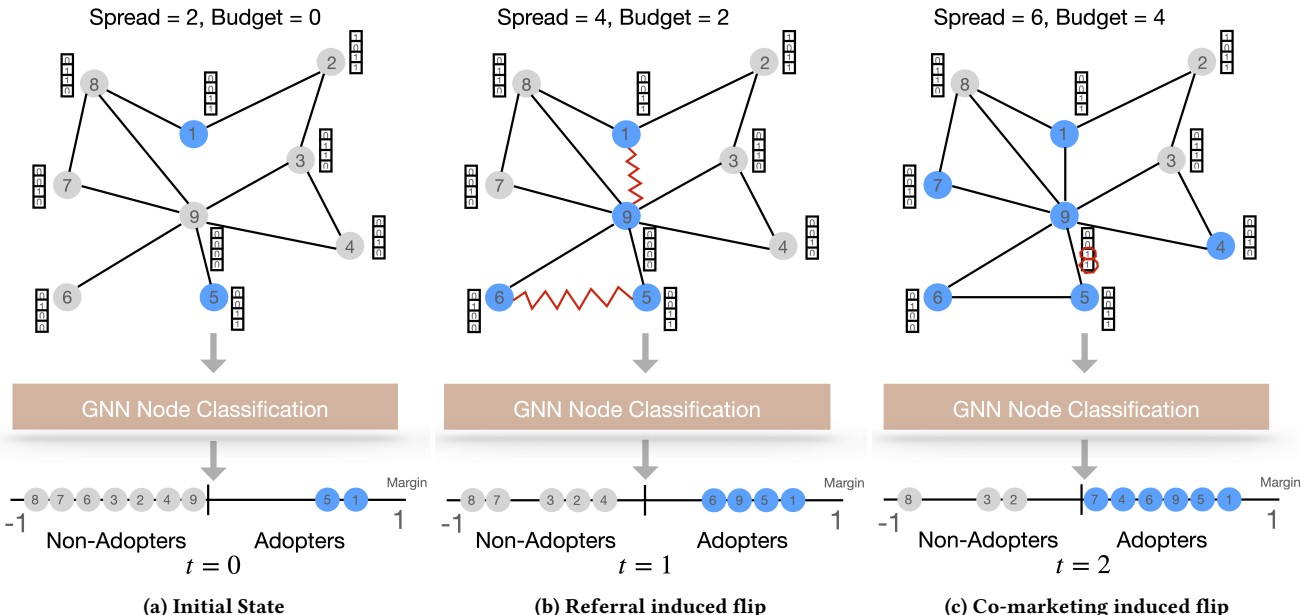

**Figure 1: Overview of the Dynamic Viral Marketing (DVM) problem. At each time step, adopters and non-adopters in an attributed user network are specified using a GNN classifier. Referral and co-marketing perturbations accelerate spreading from adopters to non-adopters. DVM seeks to find the minimum budget and dynamic perturbation set to attain a spread goal.**

the adoption labels are given by $Y^t = f(A^t, X^t, Y^0)$, where $f()$ is a general propagation model that governs the diffusion of the initial labels $Y^0$ using the network structure and attributes at time $t$. The sets of adopters and non-adopters at time $t$ are denoted by $S^t$ and $D^t$ respectively,

$$S^t = \sum_v \mathbb{1}\left[y_v^t = 1\right], \quad , D^t = \sum_v \mathbb{1}\left[y_v^t = 0\right] \quad (2)$$

The final spread, $\sigma()$ is given by the number of adopters in the network at time $T$,

$$\sigma(G^T) = |S^T| \quad (3)$$

Due to the representation learning ability of graph neural networks (GNNs) [6, 15, 17, 29] through the propagation of feature and label information, we use them as our propagation model $f()$. Specifically, a GNN $f_\theta$ is trained on the initial network $G^0$ and its parameters $\theta$ are then fixed. The GNN uses both structure and feature information to yield a decision boundary between adopters and non-adopters; the marketing objective is to flip nodes from non-adopters to adopters. Thereafter, we use *self-labeling* at time $t$: the predictions from the GNN on the network $G^t$ yield the adoption states $Y^t$. Therefore, in this framework, both our seed nodes, $Y^0$, and our propagation model, $f()$, are data-driven.

The dynamic transitions of the network from $G_{t-1}$ to $G_t$ are constrained as follows:

- *Referral marketing*: Edge insertions can be made only between $S^{t-1}$ and $D^{t-1}$ at time $t-1$; total cost of structural changes is $|A^t - A^{t-1}|$.
- *Co-marketing*: Features of nodes in $S^{t-1}$ can flip from 0 to 1 at time $t-1$; total cost of attribute changes is $|X^t - X^{t-1}|$.

As all the changes are made incrementally, the total cost incurred is given by $|A^T - A^0| + |X^T - X^0|$.

*Dynamic Viral Marketing (DVM).* : We now state the DVM optimization problem of finding the minimum budget, $\mu()$, and a minimal set of changes to reach a spread $\phi$,

$$\underset{A^1,\ldots,A^T,X^1,\ldots,X^T}{\text{arg-min}} \quad \sigma(G^T) \geq \phi \quad (4)$$

$$\mu(\phi, G_0) = |A^T - A^0| + |X^T - X^0| \quad (5)$$

Fig. 1 depicts the DVM problem schematically. Note that while the total budget is a function of the final adjacency and feature matrices, to solve the DVM problem a *sequence* of structural and attribute changes under the referral and co-marketing contraints are required. Furthermore, while we use a uniform cost model on edge and attribute perturbations, the problem can be extended to bespoke settings by using edge-specific and attribute-specific costs.

*NP-Hardness of DVM decision problem:* Consider an instance of the NP-hard *Knapsack* problem, defined by a maximum value $V$, a maximum weight $W$, and a set of $n$ items $X = \{(v_1, w_1), \ldots, (v_n, w_n)\}$ where $v_i$ and $w_i$ denote the $i^{th}$ item's value and weight respectively. The decision problem is whether there exists a subset of items $Z \subset X$ with total weight $\sum_{i \in Z} w_i \leq W$ and total value $\sum_{i \in Z} v_i \geq V$. We show that this problem reduces to the decision problem of DVM.

Given an arbitrary instance of the Knapsack problem, consider a weighted star network with target node $t$ at the center which is connected to $n$ source nodes. Each node has a single feature, whose value at node $t$ is 0 and at node $i$ is $x_i = 1 - w_i$. The weight on the edge between node $i$ and $t$ is set to $a_i = v_i$. The initial label on node $t$ is 0 and the initial label on the other nodes is 1. The cost of

**Figure 2: Overview of the Dynamic Gradient Influencing (DGI) framework. DGI picks candidate nodes to flip using Node Flipping Budget Compute, which involves gradient sorting along with bisection search, hashing and affected set estimation. The gradient-based Meta Influence Heuristic is used to tiebreak among least budget candidate nodes, as well as thresholding for Meta Attribute Flips that enhance node potency. Red lines and circles indicate candidate perturbations.**

changing the feature at a node from a value of $1 - w_i$ to 1 is $w_i$. The GNN classifier parameters are tuned such that the prediction on node $t$ flips from 0 to 1 when the weighted sum of its neighborhood of $n$ nodes with feature value $x_i = 1$ is at least $V$ (spread $\phi = 1$). The Knapsack problem is solvable iff $\sum_{i \in [n]} a_i \mathbb{1}[x_i = 1] \geq V$ and $\sum_{i \in [n]} w_i \mathbb{1}[x_i = 1] \leq W$. Thus, if the DVM problem can find feature flips costing at most $W$ at $t$'s neighbors whose edge weights sum to at least $V$ then the Knapsack problem is solvable. □

## 4.1 Relating DVM to Influence Maximization

We draw an interesting connection between DVM and the related problem of influence maximization (IM) [14]. Consider the linear threshold propagation model [10], where nodes $i \in [n]$ randomly choose a threshold $\theta_i \in [0, 1]$ and incoming influence edge weights $I_{i,j}$ such that $\forall i, \sum_j I_{i,j} \leq 1$. The propagation unfolds in discrete time steps- if the set of active nodes at any given step is $S$, then an inactive node becomes active if the following constraint is satisfied:

$$\sum_{j \in S} I_{i,j} \geq \theta_i. \tag{6}$$

While the objective in IM is to find a set of seed nodes for maximizing spread, we instead search for a sequence of dynamical changes to maximize spread. Despite the difference, given the set of adopters $S^{t-1}$ at step $t - 1$, the criterion for a node to flip in DVM has a similar form as Eq. 6. Suppose that the L-layer GNN $f_\theta$ has the associated L-step random walk propagation matrix $M$. Then the following theorem holds,

THEOREM 1. *Let the vector $\varepsilon_j^t = x_j^t - x_j^{t-1}$ denote the change in the feature of node $j$ from time $t - 1$ to $t$. Further, let the matrix $\xi = M^t - M^{t-1}$ denote the change in the L-step random walk matrix $M$ from time $t - 1$ to $t$. Then the dynamic threshold and influence edge weights for node $i$ to flip at time $t$ according to the criterion in*

*Eq. 6 are given by:*

$$\theta_i^t = z_{i,0}^{t-1} - z_{i,1}^{t-1} \tag{7}$$

$$I_{i,j}^t = M^t \alpha^T \varepsilon_j^t + \xi_{i,j}^t \alpha^T x^{t-1} \tag{8}$$

*where $\alpha$ is a vector which depends on the parameters $\theta$ of the GNN.*

The proof can be found in Sec A. Intuitively, the dynamic node threshold depends on its logit margin, and the dynamic influence edge weights depend on both the feature change $\varepsilon_j$ and the random walk propagation change $\xi_{i,j}$. Theorem 1 suggests that budget should be spent on changes which contribute most to the incoming influence weights and push the node just beyond its threshold.

## 5 Dynamic Gradient Influencing Framework

Motivated by the criterion for node flipping in Thm. 1, that the sum of the dynamic influence edge weights must exceed the dynamic node threshold, we develop the Dynamic Gradient Influencing (DGI) framework to solve the DVM problem. DGI uses Gradient-Guided Node Flipping (c.f. Sec. 5.1), to flip a particular candidate node in each step. At each step $t$, the candidate node to flip, $v^t$, is given by:

$$\underset{v \in N}{\text{arg-max}}\, I^t(v), \text{ where } N = \underset{w \in D}{\text{arg-min}}\, B^t(w) \tag{9}$$

where $B^t(w)$ denotes the budget required to flip node $w$, and $I^t(v)$ denotes the Meta Influence of node $v$. In other words, we choose the node with the least budget to flip and the highest Meta Influence. Using our novel Node Flipping Budget Computation algorithm (c.f. Sec. 5.2) candidate nodes are picked in order of lowest budget first. Further, we develop a novel Meta Influence heuristic (c.f. Sec. 5.3) for tiebreaking between equal budget candidates and thresholding for Meta Attribute Flips. We use Meta Attribute Flips to enhance a flipped node's downstream edge influence. Sec. F details DGI's

asymptotic running time complexity. The complete DGI pipeline is depicted in Fig. 2, and the algorithm can be found in Sec. C.

## 5.1 Gradient-Guided Node Flipping

In DGI, the core functionality for flipping nodes is accomplished through gradients on the restricted set of perturbations arising from the referral and co-marketing constraints of DVM. Specifically, the only changes that can happen to the adjacency matrix $A$ and feature matrix $X$ are restricted to the submatrices $A_{S,D}$ and $X_S$ respectively. Therefore, we define:

$$A_{S,D}^t = A_{S,D}^{t-1} + P_A^t \circ (\mathbf{1}\mathbf{1}^T - A_{S,D}^{t-1}), \ A_{D,S}^t = (A_{S,D}^t)^T \quad (10)$$

$$X_S^t = X_S^{t-1} + P_X^t \circ (\mathbf{1}\mathbf{1}^T - X_S^{t-1}) \quad (11)$$

where $P_A^t$ and $P_X^t$ represent the weights on the edge and feature perturbations, $S = S^{t-1}$ and $D = D^{t-1}$, and $\circ$ denotes the Hadamard product. We initialize $P_A^t = \mathbf{0}\mathbf{0}^T$ and $P_X^t = \mathbf{0}\mathbf{0}^T$.

While it is an NP-Hard combinatorial optimization problem to find the minimal perturbation that flips a node, given that the adjacency and feature matrices are both discrete, first-order gradients work well enough in practice to find the required perturbations [5, 33]. We consider the negative cross-entropy loss as our flip loss for the chosen candidate node $v \in D$:

$$L_{flip}^t(v) = \log \sigma(z_v^{t-1})_0. \quad (12)$$

Note that $y_v = 0$ for non-adopters and $y_v = 1$ for adopters. We then compute non-negative gradient scores on edge perturbation weights, $\hat{P}_A^t$, and feature perturbation weights, $\hat{P}_X^t$, with respect to the flip loss:

$$\hat{P}_A^t = \max\left(\left\lceil \frac{\partial L_{flip}^t(v)}{\partial P_A^t} \right\rceil, 0\right), \hat{P}_X^t = \max\left(\left\lceil \frac{\partial L_{flip}^t(v)}{\partial P_X^t} \right\rceil, 0\right). \quad (13)$$

We only compute the gradients once for all the perturbations. While it is possible to recompute gradients after every perturbation [5], we find that this is not that necessary to find the minimal set of perturbations. Moreover, as shown later, by merging and sorting the perturbations using the gradients, we can find the minimal perturbation set and minimum budget efficiently. Finally, suppose the minimum budget required to convert $v$ is $B(v)$, then we find the top-B(v) indices in the union of edge and feature perturbations:

$$\hat{P}^t = \text{sort}(\text{merge}(\hat{P}_A^t, \hat{P}_X^t)) \quad (14)$$

$$i_A^t, i_X^t = \text{argtop-k}_{k=B(v)} \ \hat{P}. \quad (15)$$

Using the index sets $i_A^t$ and $i_X^t$ we can make the updates to the network to convert $v$:

$$[A_{S,D}^t]_{i_A^t} \leftarrow 1, \quad A_{D,S}^t \leftarrow (A_{S,D}^t)^T \quad (16)$$

$$[X_S^t]_{i_X^t} \leftarrow 1. \quad (17)$$

## 5.2 Node Flipping Budget Computation

The budget needed to flip a node $v$ depends on both the logit margin in Eq. 7 and the node's degree $\deg(v)$ [8, 23]. In the adversarial attack framework, the practice is to set the node budget equal to its degree for local attacks [23], or choose a loss function which orders gradients in order of nodes closer to the decision boundary

for global attacks [8]. However, due to the budget minimizing objective of DVM, we need to compute the budget precisely and pick candidate nodes that need the least budget.

Therefore, to compute the minimum budget $B^t(v)$ that converts node $v$, we use the bisection method [1, 26]. For each node, we compute the sorted gradients, $\hat{P}$ in Eq. 15 once and run bisection search over these gradients to find the minimal set of perturbations required to convert $v$. We initialize the lower and upper bound for search as 0 and $\deg(v)$, the degree of $v$, respectively. Thereafter, the upper bound is doubled until it is sufficient to convert $v$. We set the maximum upper bound equal to the maximum node degree of the network. After fixing the upper bound, bisection search repeatedly halves the search interval by checking feasibility of conversion at the midpoint of the interval and converges logarithmically.

We observe that other nodes also flip due to the same structure or attribute changes made for flipping a candidate node. Therefore, to choose the best candidate, we update the budget:

$$B^t(v) \leftarrow B^t(v) + (|S^{t-1}| - |S_v^t|) \quad (18)$$

where $S_v^t$ is the set of adopters at time $t$ if node $v$ is selected as the candidate node to flip. Thus, $B^t(v)$ represents both the node flipping budget and the "collateral damage" to other nodes from flipping it. Therefore, if a node with more budget causes high collateral damage it is preferable to a node with a lesser actual budget.

Since budgets need to be recomputed for all non-adopters at every step, we make the algorithm faster by hashing node budgets and only recomputing budgets for nodes whose budget has changed. The recompute set $R^t$ is defined as the set of nodes whose logit scores changed in the previous time step:

$$R^t = \{v | v \in D^t, z_v^t \neq z_v^{t-1}\}. \quad (19)$$

For nodes whose logit scores are unchanged the actual budget might still change slightly, but for the purposes of picking the best candidate node we ignore these small changes. The entire algorithm for budget computation can be found in the the Sec. B.

## 5.3 Meta Influence Using Meta Attribute Flips

Due to the dynamic sequence of changes involved in spreading product adoption in DVM, first-order gradients in Eq. 13 are insufficient to capture the long-range effects of a perturbation. While the flipping budget is minimized at each step, we need to characterize nodes that have high influence so that adoptions can cascade sequentially. Therefore, we develop the Meta Influence heuristic to model long-range effects and estimate downstream influence. Meta Influence uses Meta Attribute Flips which are feature perturbations that increase the potency of outgoing edge perturbations at an adopter node. Consequently, the Meta Influence is defined as the normalized gradient score on an adopter's outgoing edge perturbations post Meta Attribute Flips.

For Meta Attribute Flips, we again restrict feature perturbations, this time only to the features of node $v$, and define corresponding feature perturbation weights $P_X$:

$$x_v^{t'} = x_v^t + P_X \circ (\mathbf{1} - x_v^t) \quad (20)$$

where $t'$ indicates an auxilliary time step and we initialize $P_X = \mathbf{0}$. To capture the effect of Meta Attribute Flips, we consider the following influence loss which is the sum on all non-adopters nodes

of a CW-type loss [3] that uses the logit margin, and compute gradient scores $\hat{P}_X$:

$$L_{infl}^{t'}(v) = \sum_{w \in D} (z_{w,1}^t - z_{w,0}^t) \tag{21}$$

$$\hat{P}_X^{t'} = \max \left( \left[ \frac{\partial L_{infl}^{t'}(v)}{\partial P_X^{t'}} \right], 0 \right). \tag{22}$$

Thereafter, we update the features $x_v$ using Meta Attribute Flips, which are the top-k ranked perturbations in $P_X^{t'}$, for the purposes of computing Meta Influence:

$$i_X^{t'} = \text{argtop-k} \, \hat{P}_X^{t'}, \quad [x_v^{t'}]_{i_X^{t'}} \leftarrow 1 \tag{23}$$

where $k$ is a hyper-parameter controlling the number of Meta Attribute Flips. From Thm. 1, Meta Attribute Flips find feature changes that align with the GNN's classifier weights and increase the dynamic outgoing edge influence to other nodes in Eq. 8. Further, they also help to increase the margin from the GNN decision boundary and thus increase the node's potency.

For finding the Meta Influence, we restrict the outgoing edge perturbations $P_A$ from node $v$ to the non-adopters $D$, and define corresponding edge perturbation weights $P_A$:

$$A_{v,D}^{t''} = A_{v,D}^{t'} + P_A^{t''} \circ (1 - A_{v,D}^{t'}) \tag{24}$$

where $t''$ indicates another auxilliary time step after $t'$ and $P_A^{t''} = 0$. Now consider the same influence loss in Eq. 21 on all non-adopters *at time $t''$*, and compute non-negative gradient scores $\hat{P}_A^{t''}$:

$$\hat{P}_A^{t''} = \max \left( \left[ \frac{\partial L_{infl}^{t''}(v)}{\partial P_A^{t''}} \right], 0 \right). \tag{25}$$

Note that influence loss uses the discrete perturbed feature $x_v^{t'}$, which is computed using first order gradients, therefore Meta Influence can capture second order gradient effects. Finally, we denote the Meta Influence $I^t(v)$ of a node $v$ as the normalized gradient score in $\hat{P}_A^{t''}$ averaged over the number of non-adopters:

$$I^t(v) = \frac{1^T \hat{P}_A^{t''}}{|D|}. \tag{26}$$

During candidate node selection, Meta Influence is used for tiebreaking between equal budget nodes (Eq. 9). Further, we threshold on Meta Influence to perform Meta Attribute Flips at candidate nodes after flipping:

$$x_v^t \leftarrow \begin{cases} x_v^{t'} & \text{if } I^t(v) \geq \beta \\ x_v^t & \text{otherwise} \end{cases} \tag{27}$$

where $\beta$ is a hyper-parameter controlling the threshold. Using Meta Influence, we can estimate which nodes will have high influence on their outgoing edges after Meta Attribute Flips, and thus judiciously allocate budget for Meta Attribute Flips. In Sec. 6.2, we validate that nodes with high Meta Influence indeed contribute more outgoing edge perturbations to non-adopters.

# 6 Experiments

We conducted experiments on real-world attributed network datasets to answer the following research questions:

- **RQ1**: How does the proposed DGI compare with baseline methods for DVM?
- **RQ2**: Does the proposed Meta Influence accelerate the spread and capture the node's actual dynamic influence?
- **RQ3**: What are the different kinds of cascade patterns created by DGI?
- **RQ4**: What are the properties of intermediary and susceptible nodes in cascades created by DGI?

*Datasets:* We utilize three real-world attributed datasets to evaluate DGI and baseline approaches. Epinions and Ciao [27] are datasets collected from two popular product review sites, where each user can specify their trust relation in addition to rating products. Flixster [13] is a dataset collected from a popular movie rating website with an associated social graph. We create new small-scale and large-scale splits for these datasets using the provided user networks and product ratings from [27, 34]. For small-scale split generation, we sort the nodes using their degrees and take the subgraph corresponding to the lowest degree nodes. To generate features for each user, we pick a binary value for each product/movie based on whether they rated it or not. We choose the product/movie with the least number of seeds as the optimization goal for DVM. The analysis in the main paper is conducted on the small-scale splits and their statistics are depicted in Table 1. Additional implementation details and results on large-scale splits using Fast-DGI can be found in the Sec. D.

**Table 1: Dataset statistics. $|\mathcal{S}|$ denotes seed set size.**

| Dataset | $|\mathcal{V}|$ | $|\mathcal{E}|$ | Avg, Max Deg. | $|\mathcal{S}|$ | #Features |
|---------|------|------|----------------|------|-----------|
| Flixster | 1045 | 1488 | 2.8, 153 | 5 | 839 |
| Epinions | 1054 | 1214 | 2.3, 25 | 5 | 2999 |
| Ciao | 1057 | 1190 | 2.3, 36 | 7 | 2999 |

*Evaluation metric:* The efficacy of the proposed dynamic DGI and baselines is evaluated using the minimum budget needed to spread to $C = 500$ target nodes on the respective network. We use a fixed number of targets to spread to make the results across different datasets comparable. We also report the Area Under Curve (AUC) of the budget-spread curve, which gives an aggregate estimate of the budget required for different spread values. For the purpose of calculating AUC we normalize the budget by $(\sum(A)/2 + \sum(X))$, i.e., the sum of the number of edges and turned on features. Lower values of budget and AUC indicate better performance.

*Variants:* We consider three variants of DGI for evaluation and analysis in our experiments:

- **Base** is DGI without Meta Attribute Flips. It uses Meta Influence only for tiebreaking in Eq. 9.
- **Fixed** is DGI with fixed Meta Attribute Flips. It is equivalent to using a threshold $\beta = 0$ in Eq. 7.
- **Dynamic** is DGI with optimally chosen threshold $\beta$.

**Table 2: Comparison of DGI variants to baselines. Numbers indicate minimum budget to spread to 500 nodes. GCN and SAGE are used as the GNN propagation backbones. Dynamic DGI consistently achieves the minimum budget.**

|  | Flixster | | Epinions | | Ciao | |
|---|---|---|---|---|---|---|
|  | GCN | SAGE | GCN | SAGE | GCN | SAGE |
| Degree | 2551 | 6573 | 22076 | 21125 | 25162 | 45291 |
| Margin | 8136 | 7655 | 26109 | 18550 | 20814 | 41077 |
| GradArgmax | 1012 | 625 | 5893 | 859 | 2620 | 972 |
| MiBTack | 843 | 583 | 2828 | 866 | 5111 | 1035 |
| Base | 791 | 543 | 3342 | 1099 | 3525 | 856 |
| Fixed | 831 | 571 | 1985 | 1297 | 2221 | 1094 |
| Dynamic | **667** | **494** | **1893** | **803** | **2096** | **821** |

*Baselines:* We compare with the following approaches:

- **Degree** selects target nodes based on the low-degree first heuristic. Until the target is flipped, Degree repeatedly spends a unit budget by first randomly picking a seeder node, then either adds a link from the seeder to the target if they aren't conencted, else turns on a feature with high-correlation to the label.
- **Margin** selects target nodes based on the low-margin first heuristic, i.e., nodes that have a smaller margin to the decision boundary are picked first. Edits to the structure and features are made in the same way as Degree.
- **GradArgmax** [5] is a gradient based white-box adversarial attack on structure. Target nodes are selected in the order of lower losses first. We adapt GradArgmax to make edits to both structure and features.
- **MiBTack** [34] is another white-box adversarial attack which dynamically adjusts node budgets for topology-based PGD [33]. We adapt MiBTack to make edits to both structure and features while selecting target nodes same as GradArgmax.

## 6.1 Performance Comparison (RQ1)

We compare DGI variants to baselines using budget and AUC at $C = 500$ in Table 2 and Table 3 respectively. We report results with both GCN [15] and GraphSAGE [11] as the propagation models. DGI variants outperform the baselines in all the scenarios. Among the variants, Dynamic does the best, followed by Base and then Fixed. Fixed overspends budget on Meta Attribute Flips over Base by using a threshold of $\beta = 0$, and Dynamic spends the least budget by optimally selecting nodes with high Meta Influence (for spending additional budget) for Meta Attribute Flips. Further, we plot budget spread curves on Flixster and Epinions in Fig. 3. Dynamic DGI consistently achieves the minimum budget across all levels. To understand time evolution of the spread for different variants, we plot spread as a function of time in Fig. 4a. Due to the accelerating effect of Meta Attribute Flips, Fixed spreads fastest, followed by the economical Dynamic and the conservative Base approach.

## 6.2 Strategy of Meta Influence (RQ2)

To validate the effectiveness of Meta Influence and Meta Attribute Flips, we plot the histogram of perturbations contributed by nodes

**Table 3: Comparison of DGI variants to baselines. Numbers indicate AUC of budget-spread curves. Dynamic DGI consistently achieves the minimum AUC.**

|  | Flixster | | Epinions | | Ciao | |
|---|---|---|---|---|---|---|
|  | GCN | SAGE | GCN | SAGE | GCN | SAGE |
| Degree | 19.48 | 150.50 | 215.33 | 328.15 | 349.48 | 631.66 |
| Margin | 173.70 | 177.71 | 366.43 | 284.77 | 287.11 | 595.69 |
| GradArgmax | 23.32 | 13.72 | 112.84 | 16.36 | 54.46 | 18.52 |
| MiBTack | 21.78 | 16.77 | 50.70 | 21.03 | 80.95 | 21.91 |
| Base | 19.06 | 11.68 | 56.82 | 19.23 | 46.59 | 14.46 |
| Fixed | 20.78 | 12.53 | 36.41 | 23.75 | 36.20 | 19.82 |
| Dynamic | **18.27** | **10.17** | **31.93** | **15.18** | **33.84** | **13.33** |

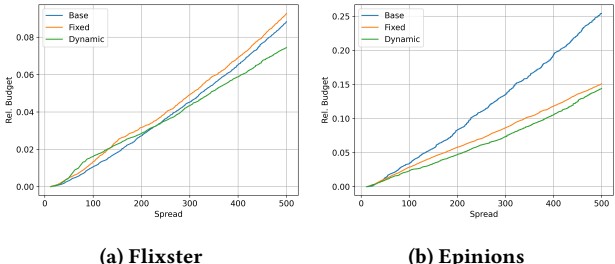

(a) Flixster          (b) Epinions

**Figure 3: Budget spent as a function of increasing spread with GCN as the GNN propagation model. Dynamic requires consistently lower budgets across all spreads.**

with respect to their Meta Influence in Fig. 4b. For each dataset, we normalize the Meta Influence to lie in the interval [0,1] and divide uniformly into 10 sub-intervals. For each sub-interval, we count the number of perturbations contributed by nodes whose Meta Influence lies within it. We carry out the analysis on Fixed DGI, in which Meta Attribute Flips are made at each node. We observe the high correlation of Meta Influence to the number of contributed perturbations across all the datasets. Note that the Meta Influence is computed at the step when the node is flipped, but even then it provides a good signal of how important that node will be later. This shows that the Meta Influence is a close approximation of the actual node influence in terms of perturbations it makes. Further, by thresholding on the Meta Influence, we are able to save budget by not applying Meta Attribute Flips on low-influence nodes.

## 6.3 Cascades created by DGI (RQ3)

To understand cascades created by the DGI spread, in Fig. 5a, we qualitatively visualize a subgraph spanned by the Dynamic DGI edges on Flixster and color each node according to its cascade hop distance from the initial seed set in this subgraph. We define the cascade hop distance of a node from the seed set inductively:

$$hop(i) = 1 + \max_{j \in PN(i)} hop(j) \quad (28)$$

where $hop(i) = 0$ for nodes in the initial seed set, and $PN(i)$ denotes perturbed neighbors of node $i$ at the step when it flips. Further, we use node size to indicate the number of perturbations the node contributes in the course of the multi-step spread. We clearly see

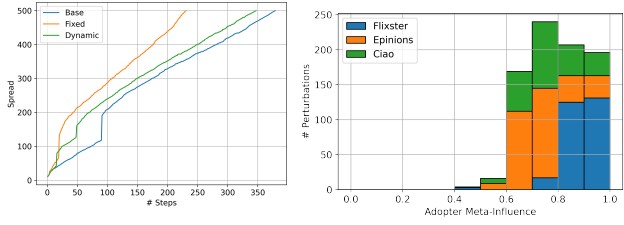

**(a) Flixster Time Evolution**  **(b) Meta Influence Strategy**

**Figure 4: (a) Spread achieved with increasing time steps for Flixster with GCN backbone. Fixed and Dynamic spread faster than Base due to acceleration from Meta Attribute Flips. (b) Histogram of perturbations contributed by intermediary adopter nodes with increasing Meta Influence, where scaling is used for mapping Meta Influence to [0,1]. Higher Meta Influence adopter nodes contribute more perturbations.**

a strong pattern of cascading flips, whereby a node flipped earlier later flips many more and so on inductively. Therefore, the DGI spread creates cascading flipping, similar to a chain of referrals in a social network, where each referral is an entirely new edge. We also see from the node sizes that a few nodes are dominant spreaders while others contribute very little.

To understand the cascades created by DGI quantitatively, in Fig. 5b we plot the number of non-adopter flips with increasing hop distances for Flixster, Epinions and Ciao. Multi-hop cascade flips account for a sizable number of the total flips, which indicates that multi-hop path flips help in decreasing the budget required for the spread. Further, the cascade hop length can be considerably large. Particularly, for Flixster, we see nodes with cascade hop lengths up to 30, indicating how the added perturbations can percolate the adoption far from the seed set.

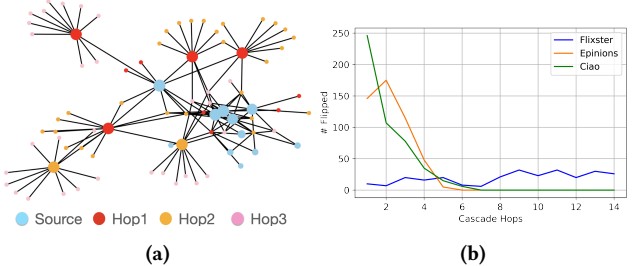

**(a)**  **(b)**

**Figure 5: (a): Visualization of cascading dynamics of DGI. Node sizes and colors correspond to number of perturbations and cascade hops respectively. Only edges added by DGI are depicted. (b): Number of flipped nodes at different cascade hops. DGI creates long and staggered cascades for DVM.**

## 6.4 Properties of Intermediary spreaders (RQ4)

To understand the properties of intermediary spreader nodes, we plot the histogram of perturbations contributed by intermediary spreader nodes with respect to their degree and classification margin in Fig. 6a and Fig. 6b respectively. For each dataset, we count

the number of perturbations arising from nodes with the degree and classification margin lying within the same sub-interval. The degree and margin are considered at the moment the perturbation is made to account for dynamic changes. Due to the degree normalization in GNN message passing, low degree nodes have a higher influence edge weight, and we observe that nodes with low degree are highly correlated to higher number of perturbations. On the other hand, high classification margin indicates high feature and neighborhood alignment with the GNN classifier, therefore making outgoing edge or feature perturbations more potent. Thus, we see that perturbations are made exclusively at nodes with the maximum possible margin.

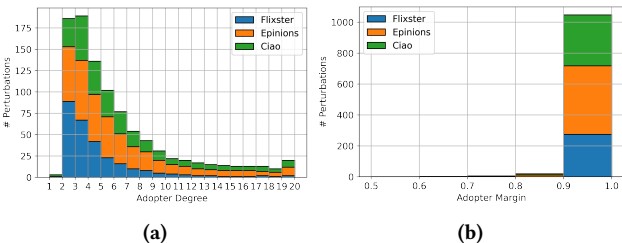

**(a)**  **(b)**

**Figure 6: (a) Histogram of perturbations contributed by intermediary spreader nodes with increasing degree. (b) Histogram of perturbations contributed by intermediary spreader nodes with increasing GCN classification margin. Higher contributions are made by spreader nodes with low degrees and high margins.**

## 7 Concluding Remarks

We proposed the novel Dynamic Viral Marketing (DVM) problem to find the minimum budget and minimal perturbation set to attain a spread goal, where the propagation model is a non-linear GNN and perturbations are restricted by referral and co-marketing constraints. We showed that DVM is NP-Hard and is related to influence maximization. We developed the Dynamic Gradient Influencing (DGI) framework, which targets non-adopters with low budget and high influence. DGI uses gradient ranking to order perturbations and utilizes an efficent budget computation approach, a novel Meta Influence heuristic, and Meta Attribute Flips to increase a node's influence. We comprehensively evaluated DGI on three real world attributed networks and demonstrated gains on average of 24% on budget and 37% on AUC over multiple gradient and non-gradient baselines. We validated the efficacy of budget computation and the Meta Influence heuristic. We extensively analyzed the cascade patterns through intermediary adopter nodes discovered by DGI.

This work opens up a new research direction, that of data-driven models for network propagation as alternatives to designed/postulated models such as Linear Threshold [10]. The data-driven models can incorporate attributes as well as long-range interactions. This research also motivates a number of future research questions including model-based reinforcement learning [21] using such GNN models in unknown environments, and the development of data-driven competitive strategies between groups each trying to increase their spread.

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

# A    Proof of Thm. 1

PROOF. Consider a L-layer GNN without non-linearities. Then the criterion to flip node $i$ at time $t$ can be expressed in terms of the L-step random walk matrix $M$ and the weights $W_l$,

$$\sum_{j \in N(i)} M_{ij}^t w_{L,1}^T \bar{W} x_j^t > \sum_{j \in N(i)} M_{ij}^t w_{L,0}^T \bar{W} x_j^t \qquad (29)$$

where $w_{L,1}$ and $w_{L,0}$ represents the final layer classifier vectors for the two classes, and $\bar{W} = \prod_{i=1}^{L-1} W_l$ is the combined feature transform of the first $L - 1$ layers.

The criterion can be equivalently written as,

$$\sum_{j \in N(i)} [M_{ij}^t w_{L,1}^T \bar{W}(x_j^t - x_j^{t-1}) + (M_{ij}^t - M_{ij}^{t-1}) w_{L,1}^T \bar{W} x_j^{t-1}$$
$$+ M_{ij}^{t-1} w_{L,1}^T \bar{W} x_j^{t-1}] >$$
$$\sum_{j \in N(i)} [M_{ij}^t w_{L,0}^T \bar{W}(x_j^t - x_j^{t-1}) + (M_{ij}^t - M_{ij}^{t-1}) w_{L,0}^T \bar{W} x_j^{t-1}$$
$$+ M_{ij}^{t-1} w_{L,0}^T \bar{W} x_j^{t-1}] \qquad (30)$$

Rearranging terms,

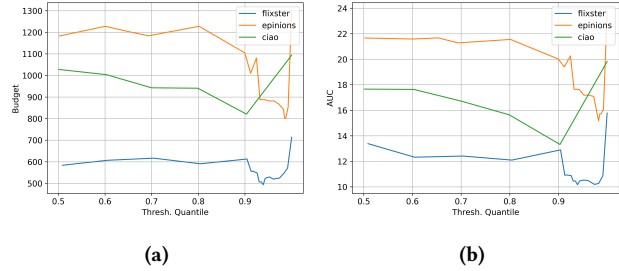

(a)            (b)

**Figure 7: Sensitivity of Dynamic DGI to the influence threshold parameter $\beta$. The x-axis represents the quantile of the threshold $\beta$ applied to the Meta Influence for each dataset. (a) represents budget and (b) represents AUC of budget-spread curve for a spread of 500.**

$$\sum_{j \in N(i)} [M_{ij}^t (w_{L,1}^T - w_{L,0}^T) \bar{W} (x_j^t - x_j^{t-1})$$

$$+ (M_{ij}^t - M_{ij}^{t-1})(w_{L,1}^T - w_{L,0}^T) \bar{W} x_j^{t-1}] >$$

$$\sum_{j \in N(i)} M_{ij}^{t-1} w_{L,0}^T \bar{W} x_j^{t-1} - \sum_{j \in N(i)} M_{ij}^{t-1} w_{L,1}^T \bar{W} x_j^{t-1} \quad (31)$$

Denoting $\alpha = \bar{W}^T (w_{L,1} - w_{L,0})$, and observing that the right hand side is the logit margin at time $t - 1$, the above simplifies to,

$$\sum_{j \in N(i)} [M_{ij}^t \alpha^T \epsilon_j^t + \xi_{i,j}^t \alpha^T x^{t-1}] > z_{i,0}^{t-1} - z_{i,1}^{t-1} \quad (32)$$

□

## B  Algorithm for Computing Budget

---
**Algorithm 1:** Bisection search to find minimum budget.

---
**Input:** $G = (A, X)$, GNN $f_\theta$, $\hat{P}$, Target node $v$.
**Output:** $B(v)$, minimum budget to flip $v$.

1  Init bounds $U = \deg(v)$ and $L = 0$
2  **do**
3    | $U = 2U$
4  **while** $v$ is not flipped by top-$U$ perturbations in $\hat{P}$;
5  **do**
6    | $C = \lfloor \frac{U+L}{2} \rfloor$
7    | **if** $v$ is not flipped by top-$C$ perturbations in $\hat{P}$ **then**
8      |  | $L = C$
9    | **else**
10   |  | $U = C$
11   | **end**
12 **while** $U - L > 1$;
  **return** : $U$

---

## C  DGI Algorithm

## D  Additional Results

### D.1  Implementation details

For the propagation model, we use 2-layer GNN architectures, as stacking multiple layers can lead to oversmoothing in GNNs [4].

---
**Algorithm 2:** DGI

---
**Input:** G=(A,X), GNN $f_\theta$, Adopters $S$, Non-Adopters $D$, $k$, $\beta$
**Output:** Minimum budget required s.t. $|S| = 500$

1  Init budget $\mathcal{B} = 0$
2  **do**
3    | **for** $v \in D$ **do**
4    |  | Compute $\hat{P}$ in Eq. 15
5    |  | Call Algorithm 1 to compute $B(v)$
6    | **end**
7    | Find $N$, the set of minimum budget nodes
8    | **for** $v \in N$ **do**
9    |  | Compute the Meta Influence $I(v)$
10   | **end**
11   | $v^* \leftarrow \max_{v \in N} I(v)$
12   | Perturb the network to flip $v^*$
13   | **if** $I(v^*) > \beta$ **then**
14   |  | Make $k$ Meta Attribute Flips at $v^*$
15   |  | $B(v^*) \leftarrow B(v^*) + k$
16   | **end**
17   | $\mathcal{B} \leftarrow \mathcal{B} + B(v^*)$
18   | $S, D \leftarrow \{v | y_v' = 1\}, \{v | y_v' = 0\}$
19 **while** $|S| < 500$;
  **return** : $\mathcal{B}$

---

**Table 4: Dataset statistics. $|\mathcal{S}|$ denotes seed set size.**

| Dataset | $|\mathcal{V}|$ | $|\mathcal{E}|$ | Avg, Max Deg. | $|\mathcal{S}|$ | Feats |
|---------|------|--------|--------------|-----|-------|
| Flixster | 3000 | 29677 | 20, 1716 | 10 | 839 |
| Epinions | 15948 | 234438 | 29, 1443 | 11 | 2999 |
| Ciao | 6841 | 77404 | 22, 749 | 6 | 2999 |

**Table 5: Comparison of DGI variants to baselines. Numbers indicate minimum budget to spread to 500 nodes. GCN and SAGE are used as the GNN propagation backbones.**

| | Flixster | | Epinions | | Ciao | |
|---|---|---|---|---|---|---|
| | GCN | SAGE | GCN | SAGE | GCN | SAGE |
| GradArgmax | 459 | 241 | 489 | 425 | 1097 | 630 |
| MiBTack | 399 | 310 | 376 | 383 | 869 | 705 |
| Base | 466 | 92 | 332 | 318 | 1136 | 615 |
| Fixed | 222 | 106 | 623 | 615 | 737 | 576 |
| Dynamic | **195** | **87** | **287** | **291** | **501** | **509** |

**Table 6: Comparison of DGI variants to baselines. Numbers indicate AUC of budget-spread curves.**

| | Flixster | | Epinions | | Ciao | |
|---|---|---|---|---|---|---|
| | GCN | SAGE | GCN | SAGE | GCN | SAGE |
| GradArgmax | 2.36 | 1.23 | 0.27 | 0.24 | 1.18 | 0.82 |
| MiBTack | 1.87 | 1.45 | 0.16 | 0.17 | 0.98 | 0.90 |
| Base | 2.38 | 0.57 | 0.13 | 0.14 | 1.20 | 0.81 |
| Fixed | 1.17 | 0.61 | 0.29 | 0.28 | 0.85 | 0.77 |
| Dynamic | **1.08** | **0.48** | **0.11** | **0.12** | **0.71** | **0.70** |

**Table 7: Hyperparameter values used for Dynamic DGI on various datasets and GNN backbones. $k$ denotes number of Meta Attribute Flips and $\beta$ (quantile) denotes the quantile of the threshold $\beta$ applied to the Meta Influence.**

|  | Flixster | | Epinions | | Ciao | |
|---|---|---|---|---|---|---|
|  | GCN | SAGE | GCN | SAGE | GCN | SAGE |
| $k$ | 2 | 2 | 4 | 2 | 4 | 1 |
| $\beta$ (quantile) | 0.75 | 0.94 | 0.65 | 0.98 | 0.30 | 0.90 |

**Table 8: Ablation study for the effect of budget compute (BC), tiebreaking (TB) using Meta Influence, and Meta Attribute Flips (MAF).**

| BC | TB | MAF | Flixster | | Epinions | | Ciao | |
|---|---|---|---|---|---|---|---|---|
|  |  |  | GCN | SAGE | GCN | SAGE | GCN | SAGE |
|  |  |  | 1012 | 625 | 5893 | 859 | 2620 | 972 |
| ✓ |  |  | 797 | 506 | 3235 | 831 | 4231 | 878 |
| ✓ | ✓ |  | 791 | 543 | 3342 | 1099 | 3525 | 856 |
| ✓ | ✓ | ✓ | **667** | **494** | **1893** | **803** | **2096** | **821** |

**Table 9: Ablation study for the effect of budget compute (BC), tiebreaking (TB) using Meta Influence, and Meta Attribute Flips (MAF).**

| BC | TB | MAF | Flixster | | Epinions | | Ciao | |
|---|---|---|---|---|---|---|---|---|
|  |  |  | GCN | SAGE | GCN | SAGE | GCN | SAGE |
|  |  |  | 23.32 | 13.72 | 112.84 | 16.36 | 54.46 | 18.52 |
| ✓ |  |  | 19.44 | 10.54 | 54.01 | 15.44 | 66.02 | 14.66 |
| ✓ | ✓ |  | 19.06 | 11.68 | 56.82 | 19.23 | 46.59 | 14.46 |
| ✓ | ✓ | ✓ | **18.27** | **10.17** | **31.93** | **15.18** | **33.84** | **13.33** |

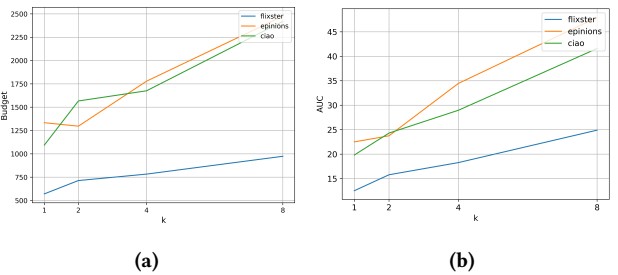

(a)          (b)

**Figure 8: Sensitivity of Dynamic DGI to the parameter $k$ controlling the number of Meta Attribute Flips. (a) represents budget and (b) represents AUC of budget-spread curve for a spread of 500.**

We report results with both GCN [15] and GraphSAGE [11] as the backbone propagation models. We set the hidden layer to size 64 and use ReLU as the intermediate non-linear function. We train models using cross entropy loss for 200 epochs, a patience of 50, learning rate 1e-2 with cosine annealing and weight decay regularization 5e-4. We use all nodes and edges during training to attain the best GNN decision boundary. All models achieve 100% accuracy on the seed set along with a small number of false positives that are in the vicinity of the seeds and are included in the initial seed set. For the spread approaches, the hyperparameter $k$ for number of Meta Attribute Flips is set to a value in $[1, 2, 4, 8, 16]$ using grid search with Fixed DGI. The threshold hyerparameter $\beta$ is set to a value within $[0, 1]$ of the maximum Meta Influence in Fixed DGI using grid search with Dynamic DGI. The maximum upper bound to convert a node is set to the maximum degree in the graph during bisection search. If the spread approach is unable to increase the size of the adopters for 40 steps, we halt and report failure. We use spread approaches with GCN backbones for all our analysis, and note that the SAGE backbone yields the same insights. DGI and baselines are implemented using the DeepRobust library in Pytorch [19] for adversarial attacks on GNNs. All GNN models and spread approaches are trained and executed on a single NVIDIA RTX2080 GPU with 8GM RAM.

## D.2 Results on large-scale data

The statistics for the larger scale datasets are presented in Table 4, and the results are presented in Table 5 and Table 6 respectively. It is interesting to note the distinction of the larger scale datasets from the smaller scale datasets in that the former are more tree like and the latter are more dense. This causes the propagation to happen much faster in the dense networks and with lesser budgets, therefore it is an easier problem when the attributed network is larger and more dense. This also matches our intuition that densely connected networks allow for faster information spread. This suggests that sparser networks require more effort for a network process such as DVM to achieve an equivalent amount of spread.

## E Ablation study

We study the effect of the different components of Dynamic DGI, i.e, budget compute (BC), tiebreaking (TB) and Meta Attribute Flips (MAF), in Table 8 and Table 9 respectively. We see that the different components indeed cause an additive increase in the performance of Dynamic DGI in all the scenarios, thus validating their utility.

## F Complexity

In each step, for each target, DGI makes 1 backward pass to compute gradients and then sorts the gradients in $O(E \log E)$ time, followed by $O(\log \Delta)$ forward passes in bisection search, where $\Delta$ is the maximum allowed budget, which we set as the maximum degree of the graph. For the set of minimum budget nodes, the Meta Influence makes two forward and two backward passes. The time taken for a forward or backward pass in a 2-layer GNN on a GPU is $O(1)$ assuming small input, hidden, and output sizes. Thus, in the first step of DGI, we make $O(|D|)$ inner steps of gradient-guided node flipping, budget computation, meta attribute flips and meta influence, which takes a total of $O(|D|(E \log E + \log \Delta))$ time. In later steps, due to budget hashing, we only recompute budgets for $O(1)$ nodes, which takes $O(E \log E + \log \Delta)$. Assuming that every step of DGI flips $O(1)$ non-adopter nodes, the total runtime complexity of DGI to achieve a spread of $\phi$ can be determined as:

$$O(\phi(E \log E + \log \Delta)) + O(|D|(E \log E + \log \Delta)). \tag{33}$$

We also use a faster variant of DGI, Fast DGI, for large scale network splits, where we recompute budgets after every $\frac{\phi}{10}$ steps, therefore bringing down the time complexity to:

$$O(E \log E + \log \Delta) + O(|D|(E \log E + \log \Delta)). \tag{34}$$

## G   Sensitivity Analysis

Table 7 reports the hyperparameters used in this work. The sensitivity of Dynamic DGI to the hyperparameter $\beta$ for threshold is presented in Fig. 7a and Fig. 7b respectively. Likewise, the sensitivity of Dynamic DGI to the hyperparameter $k$ for number of Meta Attribute Flips is presented in Fig. 8a and Fig. 8b respectively. We see that larger values of $\beta$ and smaller values of $k$ generally work better.

Received 20 February 2007; revised 12 March 2009; accepted 5 June 2009

