# OpenReview forum: "Dynamic Gradient Influencing for Viral Marketing Using Graph Neural Networks"
_ACM.org/TheWebConf/2025/Conference — WWW 2025 Poster_

### Official Review · Reviewer_oTUD · 2024-11-04

**Novelty:** 4
**Technical Quality:** 3

**Review:**

This paper presents a novel approach to viral marketing using Graph Neural Networks (GNNs) and dynamic gradient influencing. The authors propose the Dynamic Viral Marketing (DVM) problem, which aims to find the minimum budget and minimal set of dynamic topological and attribute changes to achieve a specified adoption goal. They develop the Dynamic Gradient Influencing (DGI) framework to solve the DVM problem, which targets low-budget and high-influence non-adopters and utilizes gradient ranking to find optimal perturbations.

Pros:

1.This paper introduces DVM, a novel problem using GNNs for product adoption prediction and finding optimal marketing strategies.

2.This paper develops DGI, a gradient-based framework for identifying and influencing key non-adopters in a network.

3.Comprehensive evaluation by comparing with baselines on real-world datasets demonstrates DGI’s effectiveness.

Cons:

1.How does the computational complexity of DGI scale with network size? Are there any potential bottlenecks or limitations for large-scale networks?

2.The evaluation primarily focuses on small-scale datasets, which may limit the generalizability of the findings. Exploring the performance on larger and more diverse networks would strengthen the paper.

3.How are the hyperparameters in the proposal chosen? Are there any guidelines or theoretical insights that can help determine optimal values for different datasets and networks?

**Questions:**

Please refer to the Cons listed in the review.

**Reviewer Confidence:**

2: The reviewer is willing to defend the evaluation, but it is likely that the reviewer did not understand parts of the paper

**Scope:**

3: The work is somewhat relevant to the Web and to the track, and is of narrow interest to a sub-community

---

### Official Review · Reviewer_33RE · 2024-11-30

**Novelty:** 5
**Technical Quality:** 6

**Review:**

The paper deals with the problem of viral marketing. It uses Graph Neural Networks to model the propagation of product adoption through a network. The goal is to detect customers who can be targeted with minimal budget and who will exert maximum influence over the adoption network. To this end, it introduces a dynamic component, as well as the concept of 'Meta Influence'.

I'll start by saying that I am not sufficiently familiar with the literature on the influence maximization problem, nor with the specific GNN-based techniques used here. This paper seems to be targeted at an audience that knows both very well. If the reviewer pool includes anyone who has a foot in both of these camps, their thoughts should be given greater weight.

That being said, if the goal is to reach a wider audience, this paper should try to become more easily digestible. Both the introduction and related work sections assume a great deal of pre-existing knowledge. If anything, sections 3 and 4, which lay out the notation and terminology are more easily understandable. For example, the description of the various objects, matrices, etc. in section 3 does a better job of laying out the problem structure than the introduction. It would be useful if the introduction explained these concepts at a higher level in a way that would be understandable to a marketing professional. Similarly, the concepts of referral marketing and co-marketing are mentioned in section 1, but only explained in section 4. Currently, a reader who isn't already familiar with the overall problem structure will have to jump around the paper a lot in order to get the lay of the land because the early parts of the paper are missing high-level explanations. The literature that is cited in the related work section does this more eloquently (for example, [25]), so I would recommend the authors imitate how these other papers are setting up the problem.

Other aspects of the paper that should be explained with greater clarity:
- Attributes. The paper isn't clear enough in how these work and what real-world attributes they might translate to. Some examples would help.
- "Dynamic Viral Marketing". Explain what you mean by dynamic when first mentioning it. Once again, this does become clear in section 4, but by that point you will already have lost some of your readers.
- Sections 3 and 4 need more detail on the edges of the network.
- The authors clearly put a lot of effort into Figures 1 and 2, so they deserve to be explained in much more detail.
- The paper introduces the concept of Meta Influence. This is first used in Section 5, but only actually explained in section 5.3, and even then, more detail would be good. Section 6.2 also helps with understanding, so again, it would be useful to front-load the high-level detail presented here earlier.

Other
- Figure 5 is used to illustrate the benefit of the method used in the paper, but to really do that, there would have to be another version of that plot with one of the baseline methods. This would allow for an easy visual comparison.
- Minor typo in line 220: arere

**Questions:**

- Is it realistic that the marketer knows the structure of the network?
- Are the 3 datasets used in the paper representative of what marketers would encounter in the real world?
- Fig. 1 - what are the boxes with 0s and 1s? Are they the attributes? How is the way they affect the spread illustrated in this figure? Why do nodes 6 and 9 receive the referral? Which part of the model specifies that 5 and 6 know each other so that 6 is able to receive the referral?

**Reviewer Confidence:**

2: The reviewer is willing to defend the evaluation, but it is likely that the reviewer did not understand parts of the paper

**Scope:**

3: The work is somewhat relevant to the Web and to the track, and is of narrow interest to a sub-community

---

### Official Review · Reviewer_B7fN · 2024-11-30

**Novelty:** 5
**Technical Quality:** 5

**Review:**

This paper formulates the dynamic viral marketing (DVM) problem. It first identifies that making the "dynamic" topological and attribute perturbations in network propagation has not been addressed in literature. The authors propose to utilize graph neural networks (GNNs) to model attributed network data. The GNNs can help us to test the effects of network perturbation.

Strengths:
1. The proposed GNNs can be used to identify adopter and non-adopters after the network perturbation, which alleviates the issues of data scarcity.
1. The proposed dynamic gradient influencing (DGI) framework is flexible and adaptive. The node budgets are maintained by the efficient node flipping budget computation approach. The influences are estimated by the "Meta Influence" hueristic and "Meta Attribute Flips".
1. The proposed DGI shows promising performance on three attributed networks compared to several baselines.

Concerns:
1. The motivation, background, and novelty of the work should be better illustrated in the Introduction section, especially the relationships between DVM and IM, the gradient-guided IM methods, and the details of the "meta influence" and "meta attribute flips".
1. Since the results of the perturbation in network propagation are mainly and dynamically governed by the trained GNNs, the generalizability & robustness of the model might be limited in real situations. These trained GNNs may not be able to satisfactorily replicate the reality.

**Questions:**

See Concerns 1 & 2.

**Reviewer Confidence:**

2: The reviewer is willing to defend the evaluation, but it is likely that the reviewer did not understand parts of the paper

**Scope:**

4: The work is relevant to the Web and to the track, and is of broad interest to the community

---

### Official Review · Reviewer_cWdQ · 2024-12-01

**Novelty:** 4
**Technical Quality:** 3

**Review:**

In this manuscript, the authors introduce the Dynamic Viral Marketing (DVM) problem, aimed at achieving a specified adoption goal with minimal budget and dynamic perturbations by leveraging Graph Neural Networks (GNNs) as the propagation model. They propose the Dynamic Gradient Influencing (DGI) framework, which targets low-budget, high-influence non-adopters and incorporates an efficient budget computation method alongside the Meta-Influence heuristic to assess node influence. Through comprehensive evaluations on real-world attributed networks, the proposed approach demonstrates significant performance gains, achieving a 24% reduction in budget and a 37% improvement in AUC compared to baseline methods.
However, some parts do not qualify as contributions. For example, proving that the problem is NP-Hard or its connection to influence maximization seems unnecessary. Instead, more significant aspects could be highlighted.

**Questions:**

Are three datasets sufficient for evaluation?
The comparison with previous works is limited; what is the reason?

**Reviewer Confidence:**

2: The reviewer is willing to defend the evaluation, but it is likely that the reviewer did not understand parts of the paper

**Scope:**

3: The work is somewhat relevant to the Web and to the track, and is of narrow interest to a sub-community

---

### Official Review · Reviewer_hPBW · 2024-12-03

**Novelty:** 5
**Technical Quality:** 2

**Review:**

The paper introduces the Dynamic Viral Marketing (DVM) problem, where we aim to maximize the set of adaptors (or the influence) by changes in node attributes or adding edges, over multiple time steps. The two operations are motivated by referral marketing (inserting edges) and co-marketing (changing node features). The first time step is used to train a GNN that models the influence/adaption propagation. In each subsequent timestep, we find the gradient that leads to the “flip” of a node, i.e. changes a non-adaptor to an adaptor. This seems to be motivated by literature on adversarial robustness. We then select a gradient, while considering the budget constraints, and proceed to the next round. The authors also show the NP-hardness of the problem and provide an experimental evaluation.

Modeling the influence via a GNN is not novel, and neither is the problem of maximizing the influence via network modifications. The main contribution to me is combining these aspects and finding the best modifications via a search over the gradients. This seems novel and is well-motivated, as using GNNs for estimating the influence has been successful in the past. The experimental results area also compelling, considering that there are not many benchmarks available.

There are some issues with the presentation of the paper and the definition of the problem which set back the potential contribution, and I will include these as questions. One issue is in the proof of NP-completeness of the problem. The proof relies simply on the hardness of the knapsack problem, and the reduction is simply done by adding edge weights. The original problem formulation does, however, not consider weights. I believe that even without weights one could show NP-completeness, and the proof would be more interesting.

**Questions:**

Questions

1. Why does the process proceed over multiple rounds? It does not seem like these rounds model the cascading of the influence, so which real-world process are we modeling here?
2. I don’t understand how the budget is distributed across the rounds. Is there a fixed budget per round, or an overall budget as described in (5)?
3. Can you explain how the relation to the threshold models in IM is used? Why do we now consider the random walk matrix M, and what exactly is alpha? Does it translate the GNN’s representation to a thresholding model?

**Reviewer Confidence:**

3: The reviewer is confident but not certain that the evaluation is correct

**Scope:**

4: The work is relevant to the Web and to the track, and is of broad interest to the community